# The Effects of Mono-(2-Ethylhexyl) Phthalate (MEHP) on Human Estrogen Receptor (hER) and Androgen Receptor (hAR) by YES/YAS In Vitro Assay

**DOI:** 10.3390/molecules24081558

**Published:** 2019-04-19

**Authors:** Da-Hye Kim, Chang Gyun Park, Sang Hun Kim, Young Jun Kim

**Affiliations:** 1Environmental Safety Group, Korea Institute of Science and Technology (KIST–Europe), 66123 Saarbrucken, Germany; d.kim@kist-europe.de (D.-H.K.); cg.park@kist-europe.de (C.G.P.); shkim@kist-europe.de (S.H.K.); 2Dept. of Pharmaceutical Science and Technology, Kyung Sung University, 309, Suyeong-ro, Namgu, Busan 48434, Korea; 3Division of Innovative Education and Development, Chonbuk National University, 567, Baekje-daero 54896, Korea

**Keywords:** hER, hAR, YES/YAS, MEHP, Endocrine disrupting chemicals

## Abstract

Endocrine active compounds with structural similarities to natural hormones such as 17β-estradiol (E2) and androgen are suspected to affect the human endocrine system by inducing hormone-dependent effects. This study aimed to detect the (anti-)estrogenic and (anti-)androgenic activities of mono-(2-ethylhexyl) phthalate (MEHP) by yeast estrogen/androgen bioassay (YES/YAS). In addition, the mechanism and uptake of MEHP to receptors during agonistic and antagonistic activities were investigated through the activation signal recovery test and chromatographic analysis using liquid chromatography and tandem mass spectrometry (LC-MS/MS). Estrogenic and androgenic activities of MEHP were not observed. However, MEHP exhibited anti-estrogenic (IC_50_ = 125 μM) and anti-androgenic effects (IC_50_ = 736 μM). It was confirmed that these inhibitory effects of MEHP were caused by receptor-mediated activity of the estrogen receptor and non-receptor-mediated activity of the androgen receptor in an activation signal recovery test. When IC_50_ concentrations of anti-estrogenic and androgenic activity of MEHP were exposed to yeast cells, the uptake concentration observed was 0.0562 ± 0.0252 μM and 0.143 ± 0.0486 μM by LC-MS/MS analysis.

## 1. Introduction

Phthalates are synthetic chemicals that are commonly used as plasticizers to control the rigidity of polyvinyl chloride (PVC)-based products such as medical devices and food wraps [1,2,3,4]. Phthalates are not covalently bound to plastic products and therefore, can easily leach into the environment, frequently entering the human body through direct contact or general environmental contamination [5,6,7]. Phthalate exposure has been linked to reproductive dysfunction and may affect human health [8,9,10]. While it has been shown that phthalates induce hepatotoxicity and cause hepatocellular tumorigenesis through several mechanisms [11,12], little is known about the incidence of molecular initiation events and the mechanism(s) accounting for endocrine disruption triggered by environmental stimuli. 

Mono(2-ethylhexyl) phthalate (MEHP) is a major metabolite of Di-(2-ethylhexyl) phthalate (DEHP), which is one of the most abundant phthalates and a well-known endocrine disrupting chemical (EDC) [5]. As DEHP has been widely used and is rapidly hydrolyzed to MEHP, MEHP is also widespread and detected in various environmental and human samples, including milk [13], saliva [14], human amniotic fluid [15], and blood [16,17,18]. However, MEHP has been identified as the active compound responsible for the effects of DEHP in vivo and suggested as a potential EDC, therefore, several studies have been conducted to confirm its harmful effects on the environment and humans. Among them, few studies reported the activating or inhibitory activities of MEHP to the hormone system, in particular the reproductive system. Through in vitro assay from previous studies [1,19], MEHP showed no estrogen/androgenic activities. In the case of anti-estrogenic/androgenic activities of MEHP, inconsistent results were reported. Takeuchi et al. [19] found anti-androgenic activity without anti-estrogenic activity of MEHP by CHO cell assay, while Ohtani et al. [20] revealed anti-estrogenic/androgenic activity of MEHP by two-hybrid assay. In addition, Pan et al. [4] found that urinary MEHP concentrations were inversely associated with blood testosterone levels in workers, which indicates the possibility of anti-androgenic activity of MEHP. In light of these, data regarding the effect of MEHP on the reproductive system at the cellular level are sparse, and in particular, are inconsistent. Hence, more research including new methods and approaches for a reliable assessment of MEHP are needed for understanding its impact. 

Therefore, the present study aimed to evaluate the (anti-)estrogenic and (anti-)androgenic properties of MEHP on human estrogen (hERα) and androgen receptors (hAR) and any potential adverse effects via the other mechanisms that may be relevant for further endocrine disrupting systems, with a focus on (i) measurement of agonistic and antagonistic activities of MEHP and (ii) assessment of MEHP uptake to yeast cells during agonistic and antagonistic activities. Agonistic and antagonistic activities were measured using the yeast reporter assay (XenoScreen YES/YAS assay), in which Saccharomyces cerevisiae yeast cells were exposed to serial dilutions of the MEHP in 96-well plates. In vitro yeast-based reporter assays for detecting chemicals with potential endocrine-disrupting activity have been firmly established. The main advantages of the yeast report gene assay provide the assessment for early molecular reactions caused by agonistic or antagonistic activities on hERα and hAR receptors. The observed the hormonal effects in the human YES/YAS yeast assay focused on target gene activation for receptor-ligand interaction and the hormonal action could be different with several types of the cell due to their additional signaling pathway and phylogenetic diversity. Consequently, human hormonal-based yeast bioassays are more suitable to identify substances that could influence the potential human endocrine system. Therefore, additional in vitro bioassays based on yeast cells for (anti)estrogens and (anti)androgens were validated for our studies. Besides, this assay is more tolerant on cytotoxic effects, Hence, it implements with reliable results for receptor-ligand interaction even high concentration conditions [5,21]. The uptake of MEHP in yeast cells was confirmed by liquid chromatography and tandem mass spectrometry (LC-MS/MS).

## 2. Results 

### 2.1. Estrogenic and Androgenic Activity

In this study, the possibility that MEHP can interact with hERα and hAR as an agonist was assessed. Before confirmation of the estrogenic and androgenic activity of MEHP, the effect of MEHP on yeast cell viability was measured (Figure 1). The cytotoxicity testing revealed no effect at concentrations ranging from 100 nM to 1.00 mM, however, significant cytotoxicity of MEHP was observed in samples which exposed 10.0 mM of MEHP. This result was similar to the latest study which showed the cytotoxicity was induced at 1.60 mM of MEHP [22]. Thus, the maximum exposure concentration of MEHP for YES and YAS assays was set at 1.00 mM.

The E2 equivalents in the YES test were calculated using a seven-point standard E2 curve (10.0 pM to 10.0 nM) for confirmation of estrogenic activity of MEHP. The EC_50_ value for E2 was determined to be in the sub-nanomolar range, as shown in Table 1 (EC_50_ = 1.26 nM). The degree of matching of the sigmoidal dose-response function was variable and only observed for E2 (R^2^ = 0.9842). Figure 2A shows a graphical relationship of percentage induction of gene synthesis and concentration of E2 and MEHP. Compared with E2, MEHP stimulated less gene expression of β-galactosidase (<3% of induction). In the case of androgenic activity, the 5α-dihydrotestosterone (DHT) equivalents in the YAS test were calculated using a seven-point standard DHT curve (1.00 nM to 1.00 μM). The EC_50_ value for DHP was determined to be in the nanomolar range, as shown in Table 1 (EC_50_ = 16.3 nM). The degree of matching of the sigmoidal dose-response function was variable and only observed for DHT (R^2^ = 0.9496). Figure 2B shows a graphical relationship of percentage induction of gene synthesis and concentrations of DHT and MEHP stimulated β-galactosidase gene expression (<2% of induction). As Figure 2B, it is observed that MEHP does not act similar to E2 and DHT and induce estrogenic and androgenic activities as an agonist.

### 2.2. Anti-Estrogenic and Anti-Androgenic Activity

For confirmation of anti-estrogenic activity, the medium with 1.00 nM of E2 (negative control) was regarded as producing 100% induction of estrogenic activity. The positive control was 4-hydroxytamoxifen (HT), a series of seven concentrations (10.0 nM to 10.0 μM). MEHP treated a series of eight concentrations (316 nM to 1.00 mM) to yeast cells, considering cytotoxicity results of this study. In this test, HT and MEHP showed inhibition activity as a dose-response curve. The IC_50_ values of HT and MEHP were 1.11 μM and 125 μM, respectively, as shown in Table 2. In addition, the degrees of matching were observed for HT and MEHP (R^2^ = 0.8546 and 0.9068, respectively). Figure 3A represents a graphical relationship that was expressed as a percentage of the induction ratio according to concentrations of MEHP and HT. MEHP inhibited the production of receptor-mediated β-galactosidase by E2 at least 50% and maximal inhibition was shown at the highest assessed concentrations (13.4% induction at 1.00 mM) (Figure 3A). 

For confirmation of anti-androgenic activity of MEHP, medium with 31.6 nM of DHT was used as negative control, and it was regarded as producing 100% induction of androgenic activity. The positive control was flutamide (FL) and treated a series of seven concentrations (0.10 μM to 0.10 mM). MEHP treated a series of eight concentrations to yeast cells. In this test, FL and MEHP showed inhibitory activity. The IC_50_ value was 20.3 μM for FL, as shown in Table 2. The IC_50_ value of MEHP was 736 μM, and it was higher than the IC_50_ value for anti-estrogenic activity (Table 2). The degrees of matching of sigmoidal dose-response functions were variable, and degrees of matching were observed for FL and MEHP (R^2^ = 0.8654 and 0.7665, respectively). Figure 3B shows a graphical relationship expressed as a percentage of the induction ratio according to concentrations of MEHP and FL. MEHP only inhibited the production of β-galactosidase from 100 μM concentration, and maximal inhibition of MEHP was at the highest assessed concentrations (45.0% induction at 1.00 mM).

Results of the signal activation recovery test for MEHP in YES and YAS assay are presented in Figure 4. Generally, the antagonist activity of compounds can be divided into largely specific effects or non-specific effects. In YES and YAS assays, the antagonist activity might be caused by non-specific inhibition of induction, not as a specific effect due to non-receptor mechanisms, including via protein synthesis inhibition, disruption of transcription of the β-galactosidase report gene resulting in a decrease of receptor binding. Therefore, the recovery assay was performed to identify a decrease in induction of β-galactosidase by ER or AR receptor-mediated mechanisms. Each agonist of different concentrations was considered as a negative control (producing 100% induction of β-galactosidase, respectively). With the increasing concentration of the E2 agonist, as shown in Figure 4A, the increasing dose-response inhibition activity for MEHP was observed. Treatments with the maximum concentration of E2 as an agonist (1.00 nM) showed the highest linear regression degrees among other agonist concentrations. 

### 2.3. Uptake of MEHP in Yeast Cells 

Through LC-MS/MS analysis, the uptake concentration of MEHP in yeast cells was investigated to confirm the actual inhibitory concentration of MEHP to normal estrogen and androgen activities. Two concentrations of MEHP (125 μM and 736 μM as IC_50_ value of anti- estrogen and -androgen activities) were treated to YES and YAS yeast cells. As result, concentrations of MEHP (mean ± standard deviation) were 0.0562 ± 0.0252 μM in YES yeast and 0.143 ± 0.0486 μM in YAS yeast. These results indicate that the MEHP uptake to yeast cells is very low and actural inhibitory concentration of MEHP is also much lower than treated concentrations of MEHP. In addition, our result of MEHP uptake is suggested that the actual IC_50_ value of anti-estrogen and anti-androgen activities of MEHP in this study are 0.0562 μM and 0.143 μM. There is one previous study reported IC_50_ value of anti-androgen activity of MEHP [20] and our result is much lower than them (0.65 μM). 

## 3. Discussion

The previous studies on the estrogenic activity of MEHP in vitro assays have been conducted. Similar to our result, most of the previous studies reported none estrogenic activity of MEHP [1,19,23,24]. Only one study reported the estrogenic activity of MEHP [25]. They used embryonic leydig cell and MCF-7 cell and 10 μM of MEHP showed the estrogenic effect depending on the transfection of estrogen responsive elements (ERE-luc or AP-1-luc) [25]. It concluded that the estrogenic agonist was caused by estrogen responsive elements, not as estrogen receptors. In the case of androgenic activity, some studies were conducted and reported similar result with our results. Takeuchi et al. [19] used hamster ovary cells (CHO-K1 cell) and reported no androgenic activity of MEHP. And another research used a computational network model and reported no agonistic effect of MEHP [26]. In addition, Stroheker et al. [27] confirmed that MEHP did not affect testosterone production on fetal testis. In light of these results, it is suggested that MEHP has no agonistic effect on estrogen and androgen hormone receptor. Regarding the anti-estrogenic activity of MEHP, some previous studies were performed, but inconsistent results were reported. One study confirmed the decrease of natural estrogenic activity on MCF-7 cells due to MEHP exposure [28]. In addition, Ohtani et al. [20] investigated the anti-estrogenic activity of MEHP using a two-hybrid assay in which yeast was integrated with the human estrogenic receptor. Similar to our study, they confirmed that MEHP showed anti-estrogenic activity at the human estrogenic receptor, and it was expressed as relative activity within 67.0 μM of MEHP in 300 pM of E2 as the agonist. On the other hand, recent study about assessing the relation between MEHP and estrogen receptor using the computational network model showed no anti-estrogenic activity of MEHP [23]. Another previous study also revealed no anti-estrogenic activity of MEHP through the measurement with CHO cells that were transiently transfected with an expression plasmid for human ERα or ERβ receptors [19]. However, they were exposed to 10.0 μM of MEHP in the presence of 10.0 pM E2 for hERα or 100 pM E2 for hERβ, and were similar to our result that anti-estrogenic activity was not observed when cells were exposed to 10.0 μM or less of MEHP. In fact, it is difficult to compare accurately the results of these previous studies due to the difference in the treated E2 concentration as the agonist and also the most of performed in vitro tests were applied on mammalian or human cells. These cells could be more sensitive on cytotoxicity than yeast cell and this cytotoxicity might affect the antagonist assessment [19,23,28]. Nevertheless, in light of the previous and present studies, it seems that MEHP has a potential inhibitory effects on estrogenic activity.

Thus far, some in vitro and in vivo studies have been conducted on the anti-androgenic activity of MEHP. Unlikely our result, none of the in vitro studies reported inhibition of MEHP on normal androgenic activity [19,26,27,29]. However, in in vivo study, the anti-androgenic effect of MEHP was reported through dosing experiment to male rats. They confirmed the decreased weights of ventral prostate, seminal vesicles and levator ani/bulbocavernosus (LABC) [30]. Additionally, another human study examined the relationship between blood testosterone levels and urinary MEHP concentrations of workers at a factory producing unformed polyvinyl chloride flooring who were exposed to DEHP [4]. They revealed that exposed workers with high urinary MEHP concentrations (over 10.0 μM) showed significantly lower blood testosterone levels than unexposed workers. In addition, they showed that the testosterone level decreased significantly with increasing total phthalate ester, including MEHP, through regression analysis [4]. Thus, we assumed that secretion of testosterone does not properly occur with exposure to high concentrations of MEHP due to its inhibition of normal androgen activity inhibition. In line with the results of previous and current studies, it is speculated that MEHP does not react as an antagonist on the androgen receptor, but the anti-androgenic activity could be occurred by other mechanisms when MEHP was exposed to high concentrations. Thus, we performed the signal activation recovery test for understanding the antagonist mechanism of MEHP.

Lee and Guven et al. observed higher inhibition rates with increasing concentrations of the agonist compounds. This finding suggests the direct receptor-mediated action of the antagonist compound, and not toxic or non-specific activity [21,31,32]. In this study, the addition of three E2 agonist concentrations were recovered with signal dose-responses by MEHP exposure, which were active in the hERα antagonism assay. These results confirmed that the anti-estrogenic activity was due to receptor-mediated mechanisms. On the other hand, as shown in Figure 4B, with the increasing concentration of the DHP agonist, MEHP showed the similar inhibition activities and regression degrees, unlike with the recovery test of the E2 agonist. Thus, the anti-androgenic activity of MEHP was likely related to non-receptor mechanisms, such as protein synthesis inhibition, disruption of β-galactosidase gene transcription, or enzyme inhibition. These non-receptor mechanisms could result in a decrease of induction by high concentrations of MEHP. Therefore, additional studies are needed for understanding the main cause of the anti-androgenic activity among non-receptor mechanisms (i.e., protein synthesis inhibition, disruption of β-galactosidase gene transcription, or enzyme inhibition).

## 4. Materials and Methods 

### 4.1. Chemicals and Reagents

MEHP (97.0% pure, CAS # 4376-20-9) was purchased from Sigma-Aldrich (New Haven, CT, USA). ^13^C_2_-labeled MEHP (CLM-4584-MT-1.2, Cambridge Isotope Laboratories, Tewksbury, MA, USA) were used as the internal standard of MEHP. The XenoScreen YES/YAS assay kit was obtained from Xenometrix AG (Allschwil, BL, Switzerland). Acetonitrile and methanol, LC-MS grade, were purchased from VWR (Leuven, VB, Belgium). Formic acid, ammonium acetate and dimethyl sulfoxide (DMSO) were provided from Sigma-Aldrich. MEHP solutions were prepared in DMSO, and treated DMSO did not exceed 1% (*v*/*v*), which was not cytotoxic.

### 4.2. YES and YAS Assay

In this study, yeast cells (*Saccharomyces cerevisiae*) that were genetically modified with the stably integrated human receptors hERα (YES) or hAR (YAS) were used. The yeast cells also contained an integrated expression plasmid with the reporter gene *lac-*Z, which encodes the enzyme β-galactosidase. The activation of hERα and hAR by the binding of an estrogen and androgen active substance led to the expression of β-galactosidase, which converted the substrate chlorophenol red-β-d-galactopyranoside (CPRG) into chlorophenol red. The yeast growth and color changes were calculated by UV/Vis spectroscopy (TECAN, Männedorf, ZH, Switzerland) at 690 and 570 nm wavelength-light, respectively. The results were estimated in terms of agonistic and antagonistic effects and were also evaluated for cytotoxic effects by testing the optical density of each well (λ = 690 nm). 

For the evaluation of the agonist and antagonist activity of MEHP, eight different dilution concentrations (316 nM–1.00 mM, half-log dilutions) were determined by the dose-response curve of cell cytotoxicity. For the reference of the agonist test, E2 and DHT standards were used for calibration curves, and for the antagonist test, non-saturating concentrations of E2 (1.00 μM) and DHT (31.6 μM) were spiked with MEHP in the test medium. HT and FL were used as positive controls for the antagonist test, respectively. The treated concentrations of positive and negative controls were followed by protocol of YES/YAS assay. All of the chemicals, including MEHP, were dissolved in DMSO, and each standard compound was diluted at seven concentrations (half-log dilutions). Each assay was repeated three times. Moreover, both yeast cells were assessed for activation signal recovery in order to understand the mechanism of the antagonistic effect of MEHP on both receptors because non-receptor-mediated mechanisms might induce antagonistic effects. For recovery, MEHP solutions were incubated with three different amounts of an agonist (E2 in YES test: 0.10, 0.32, and 1.00 nM; DHT in YAS test: 3.16, 10.0, and 31.6 nM). Three concentrations of MEHP (1/10 of the IC_50_ value, IC_50_ value, and 10-fold IC_50_ value) were tested.

### 4.3. Uptake of MEHP in Yeast Cells

#### 4.3.1. Pretreatment

The initial process of pretreatment is same with YES/YAS assay, but treated MEHP concentration was IC_50_ concentrations of MEHP which determined in assays for anti-estrogenic and -androgenic activity (125 μM and 736 μM) and CPRG was not added. After incubation for 48 h, the samples were transferred to a microtube and centrifuged at 6000× *g* for 5 min in order to discard the supernatants. The pellets were washed by medium and were centrifuged again. Then, 200 μL of medium was added to each sample with mixing and the yeast cell numbers were counted. The glass beads were added into the samples for breaking cell membrane and each sample was vortexed with 10 times for 30 s. After that, the yeast cell numbers were counted again to confirm the number of breaking cells. Each 100 μL of lysates was obtained after centrifugation and spiked with 20 ng of MEHP internal standard. The samples were diluted using methanol more than 20-fold to reduce the matrix effect and avoid the LC-MS/MS contamination. Subsequently, the samples were filtered using a 0.2 μm of nylon filter before analysis of LC-MS/MS. 

#### 4.3.2. LC-MS/MS Analysis

The analysis of samples was conducted by high performance liquid chromatography-tandem mass spectrometry using an Agilent 1200 HPLC system with a 6460 electrospray triple-quadrupole mass spectrometer (Agilent Technologies, Santa Clara, CA, USA). The mobile phases were 0.1% acetic acid in acetonitrile (A) and 0.1% acetic acid in water (B). To separate MEHP from the extracts, a ZORBAX Eclipse Plus C18 column (2.1 mm × 50 mm, 1.8 μm) fitted with a ZORBAX Eclipse Plus C18 guard column (2.1 mm × 5 mm, 1.8 μm) were used. The mass spectrometer (Agilent Technologies, Santa Clara, CA, USA) was operated in electrospray negative ionization mode, and the identification and quantification of MEHP in samples was achieved in multiple reaction monitoring (MRM) mode (Table 3) by a 95:5 ratio of mobile phase solvents.

### 4.4. Data Analysis

Cytotoxic effect was expressed through growth factors G (Equation (1)), and growth factor of ≤ 0.5 was considered to be toxic effect.
(1)$$G\;=\;\frac{A_{690,S}}{A_{690,N}}$$
where $A_{690,S}\;$ is the absorbance of the sample exposed to MEHP at 690 nm and $A_{690,N}$ is the absorbance of the solvent control at 690 nm. The agonist and antagonist data were expressed as the means ± SD (standard deviation). Dose–response curves were completed using the sigmoid dose–response function in Sigmaplot software (version 1.25, San Jose, CA, USA) and expressed as a relative percentage of induction based on each induction ratio (I_R_) [5,21]. The detailed calculation steps were as follows: (1) Calculate the delta OD_570_–OD_690_ of wells. (2) Calculate the mean values of the standards and samples. (3) Calculate growth factor G (absorbance of sample at 690 nm/absorbance of solvent control at 690 nm), β-galactosidase activity U_s_ (absorbance of sample at 570 nm/absorbance of sample at 690 nm), and induction ratio I_R_ (Equation (2)). (4) Draw dose-response curves and determine the activities of MEHP.

(2)$$I_{R}=\frac{1}{G}\;\times \;\frac{A_{570,S}\;(\mathrm{net}\;\mathrm{absorbance}\;\mathrm{of}\;\mathrm{the}\;\mathrm{sample}\;S\;\mathrm{at}\;570\;\mathrm{nm}-690\;\mathrm{nm})}{A_{570,N}\;(\mathrm{net}\;\mathrm{absorbance}\;\mathrm{of}\;\mathrm{the}\;\mathrm{solvent}\;\mathrm{control}\;\mathrm{at}\;570\;\mathrm{nm}-690\;\mathrm{nm})}$$

The EC_50_ and IC_50_ values of compounds that showed dose-response curves were calculated (Table 1 and Table 2). When the induction ratio was over 10 percent of the difference between the maximum E2 or DHT response and solvent control, the MEHP was regarded as an agonist. If exposure to MEHP inhibited E2 (YES) or DHT (YAS) agonists at least 50 percent in medium (negative control), it was regarded as antagonist.

## 5. Conclusions

In this study, agonistic/antagonistic activities of MEHP on human estrogen and androgen receptors were investigated using YES/YAS assay, the activation signal recovery test and LC-MS/MS analysis. MEHP did not induce yeast toxicity and agonistic activities at concentrations up to 1.00 mM, and MEHP exposure showed a significant antagonist effect with reliable sigmoidal functions. The IC_50_ values were estimated as 125 μM for anti-estrogenic activity and 736 μM for anti-androgenic activity by yeast assay. However, actual uptake of MEHP to yeast cells were confirmed as 0.0562 ± 0.0252 μM and 0.143 ± 0.0486 μM when the MEHP concentration in samples exposed 125 μM and 736 μM of MEHP as IC _50_ from yeast assay by LC-MS/MS analysis. This result is suggested that actual IC_50_ of anti-estrogenic and -androgenic activities of MEHP is much lower than estimated IC_50_ by yeast assays. The activation signal recovery test confirmed that the anti-estrogenic effect of MEHP was caused by direct-receptor activity, whereas the anti-androgenic effect of MEHP appears to be related to non-receptor-mediated mechanisms such as protein synthesis inhibition, disruption of β-galactosidase gene transcription, and enzyme inhibition at high concentrations. In light of these results from the YES/YAS assay and the activation signal recovery test, it is speculated that MEHP can induce anti-estrogen and anti-androgen activities by directly binding with receptors or via non-receptor-mediated actions.

## Figures and Tables

**Figure 1 molecules-24-01558-f001:**
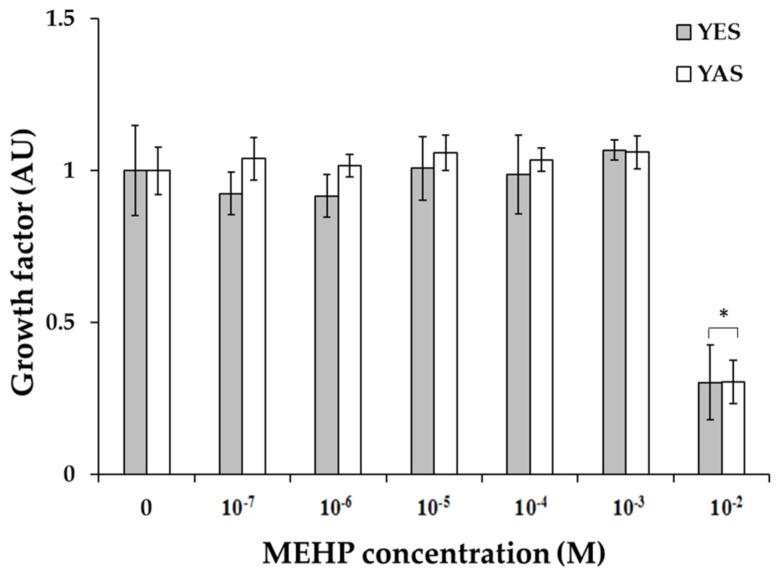
The growth factors of YES and YAS Saccharomyces cerevisiae yeasts after MEHP exposure. Asterisk means the growth factor of samples is below 0.5.

**Figure 2 molecules-24-01558-f002:**
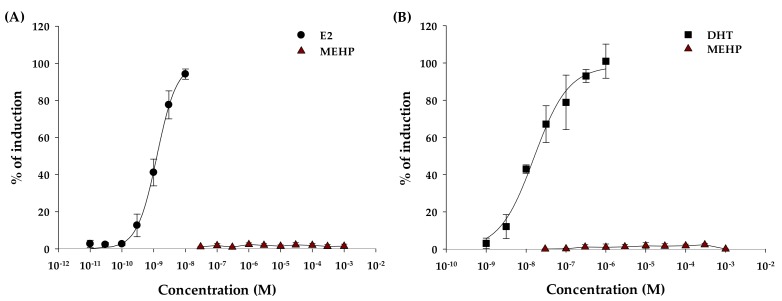
Sigmoidal dose-response curves for the agonistic potential of MEHP on estrogen (**A**) and androgen (**B**) receptor activity, respectively.

**Figure 3 molecules-24-01558-f003:**
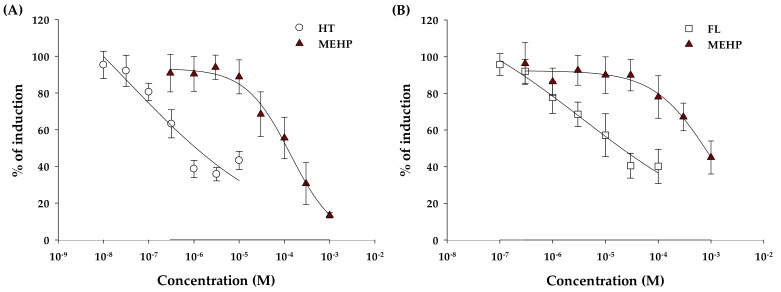
Sigmoidal dose-response curves for antagonistic potential of MEHP to estrogen (**A**) and androgen (**B**) receptor activity, respectively.

**Figure 4 molecules-24-01558-f004:**
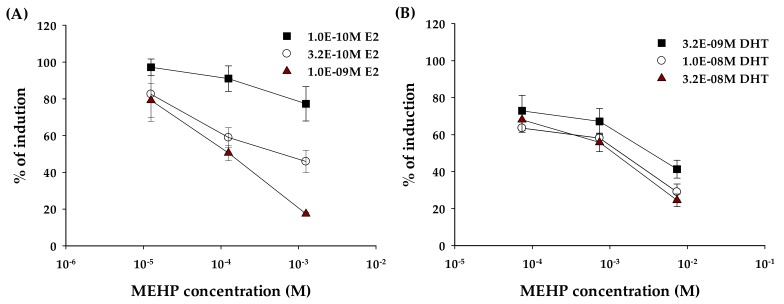
The antagonistic effects of MEHP according to the different concentrations of estrogenic (**A**) and androgenic (**B**) activity, respectively.

**Table 1 molecules-24-01558-t001:** EC_50_ values with 95% confidence intervals and R^2^ for agonist standards and MEHP.

Agonists	EC_50_ (nM)	95% Confidence Interval (nM)	R^2^
E2	1.26	1.16–1.39	0.9842
MEHP (ER) ^a^	-	-	-
DHT	16.3	13.1–20.5	0.9496
MEHP (AR) ^a^	-	-	-

^a^ Where compound did not produce over 10 percent of the difference between the maximum E2 or DHT response and solvent control.

**Table 2 molecules-24-01558-t002:** IC_50_ values with 95% confidence intervals and R^2^ for antagonist standards and Mono-(2-EthylHexyl) Phthalate (MEHP).

Antagonist	IC_50_ (μM)	95% Confidence Interval (μM)	R^2^
HT	1.11	0.73–1.73	0.8546
MEHP (ER)	125	90.0–151	0.9068
FL	20.3	13.6–32.2	0.8654
MEHP (AR)	736	604–1740	0.7665

**Table 3 molecules-24-01558-t003:** MRM transition and retention time of MEHP.

	RT	Precursor Ion (*m*/*z*)	Product Ion (*m*/*z*)
MEHP	5.765	277	134.2, 127.2
^13^C_2_-MEHP	5.762	281	137.0, 79.1

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
