# Peer review of "The Effects of Mono-(2-Ethylhexyl) Phthalate (MEHP) on Human Estrogen Receptor (hER) and Androgen Receptor (hAR) by YES/YAS In Vitro Assay"

_molecules, 2019, doi:10.3390/molecules24081558_

Round 1

Reviewer 1 Report

Reviewer’s criticism to MOLECULES-492215 manuscript

In this study, authors investigated the estrogen/androgen and anti-estrogen/androgen activities of mono–(2–ethylhexyl) phthalate (MEHP). The whole manuscript is well described and its structure is traceable.

Therefore, I can recommend this paper for publication in Molecules but I have some comments and so minor revision is needed.

Detailed comments

1; Page 5 Line 149-Page 7, Line 206

Based on this part, the mention of “non-receptor-mediated activity of the androgen receptor” in Abstract and Conclusion section should be considered regarding the mechanism of anti-androgenic activity (exposed to high concentrations) what is currently unknown.

2; Page 7, Line 216-220

To the appropriate evaluation and discussion of these results, the possible effects of the sample preparation and the losses during the extraction should be considered. Please mention this in this part.

3; Page 9, Line 272-281

Was the diluted and filtered medium (which contains the breaking cells) injected without any other sample preparation?

What was the composition of the used medium?

Minor suggestions

Page 1, Line 28 Keywords

Endocrine disrupter chemical should be changed Endocrine disrupting chemicals

Page 2, Line 45-46

Use the ED abbreviation

Based on these critical issues, reviewer can recommend the manuscript for publication in the Molecules but minor revision is needed.

Author Response

Dear reviewer,

Thank you very much for your comments and suggestions. The comments and suggestions are valuable and very helpful for revising and improving our manuscript.

Based on this part, the mention of “non-receptor-mediated activity of the androgen receptor” in Abstract and Conclusion section should be considered regarding the mechanism of anti-androgenic activity (exposed to high concentrations) what is currently unknown.

>> Response:  Thank you for your comment. we added the sentence in manuscript which is red marked. 

In non-receptor-mediated activity of the androgen receptor can be explained by protein synthesis inhibition, disruption of β-galactosidase gene transcription, enzyme inhibition and etc.

To the appropriate evaluation and discussion of these results, the possible effects of the sample preparation and the losses during the extraction should be considered. Please mention this in this part.

>> Response: To confirm the MEHP uptake to yeast cells, we used internal standard method using labeled MEHP standard (13C2-MEHP) from Cambridge isotope laboratories. Through the internal standard method, the loss of analyte during sample preparation or sample inlet. We have mentioned this information in revised manuscript which is red marked.

Was the diluted and filtered medium (which contains the breaking cells) injected without any other sample preparation?

What was the composition of the used medium?

>> Response: We removed the breaking cells from samples through centrifugation and then diluted samples using methanol (more than 20-fold) to reduce the matrix effect and avoid the LC-MS/MS contamination. We have added these detailed descriptions in revised manuscript which is red marked   

Comment 4: Page 1, Line 28 Keywords

Endocrine disrupter chemical should be changed Endocrine disrupting chemicals

>> Response: Sorry for our mistake. We have revised it.

Comment 5: Page 2, Line 45-46

Use the ED abbreviation

>> Response: As reviewer’s pointed out, we have revised to EDC which is red marked.

Reviewer 2 Report

In this study, the authors examined the hormonal action of MEHP by using YES/YAS in vitro assay. The experimental procedure is considered to be well established by the authors. The authors provided useful information for the research in endocrinology and environmental science.

As you know, the steroid hormonal action of MEHP is being established in several types of cells. Therefore, the authors should clearly describe about the benefits of using YES/YAS in vitro assay for screening of endocrine disrupting chemicals.

Results and discussion should be in separate section, as original results in this research and previous reports are confused.

Author Response

We would like to thank the detailed two comments, whose comments significantly improved our manuscript.

As you know, the steroid hormonal action of MEHP is being established in several types of cells. Therefore, the authors should clearly describe about the benefits of using YES/YAS in vitro assay for screening of endocrine disrupting chemicals.

- We added the description for yes/yas benfits as follow,

The observed the hormonal effects in the human YES/YAS yeast assay focused on target gene activation for receptor-ligand interaction and the hormonal action could be different with several types of the cell due to their additional signaling pathway and phylogenetic diversity. Consequently, human hormonal-based yeast bioassays are more suitable to identify substances that could influence the potential human endocrine system. Therefore, additional in vitro bioassays based on yeast cells for (anti)estrogens and (anti)androgens were validated for our studies.

Results and discussion should be in separate section, as original results in this research and previous reports are confused.

-We agree with your comments. We divided into two sections according to your suggestion.

Reviewer 3 Report

My initial comments were (1) this paper confirmed the lack of any significant activity of MEHP for ER or AR and(2) the authors did not cite a large body of relevant literature.As far as I understand, the mechanism for the "phthalate effect" on the developing testes is not understood, and is not due to direct ER or AR activity. The revised paper still confirms that, and maybe publishing negative results is OK. The authors have now covered the relevant literature 

Author Response

We would like to thank the reviewer for their detailed comments and suggestions for the manuscript. We are very encouraged that the your positive comments for our negative results for hormonal effects on MEHP.

This manuscript is a resubmission of an earlier submission. The following is a list of the peer review reports and author responses from that submission.

Round 1

Reviewer 1 Report

This manuscript describes experiments on the in vitro effect of MEHP as an agonist and antagonist on the human estrogen and androgen receptors (ER and AR). The authors make claims that this is novel work because there is little in vitro data for MEHP and ER/AR. My main criticism is that there is significant data in the open literature, listed here. Most to all of this has concluded that MEHP is inactive against ER and AR. The authors here have simply recapitulated this negative result except for finding some potential for ER antagonist activity at very high concentrations (>100 uM). Over a range of cell types, MEHP shows cytotoxicity starting at ~6 uM with most cell types being cytotoxic by 50 uM, which would confound the antagonist results (which is what is being seen in the high dose AR antagonist results. See the following paper for information on cell-stress cytotoxicity interference with in vitro assays in general https://academic.oup.com/toxsci/article/152/2/323/2578946 and the following for specific information on interference with AR antagonist assays https://pubs.acs.org/doi/abs/10.1021/acs.chemrestox.6b00347

Even if there was ER antagonist activity at 125 uM, the authors would need to make a case that it is of any environmental concern, but finding some exposure scenario that would result in such high concentrations in vivo. Looking at the cited references, concentrations of 0.1 uM in breast milk was seen (ref 13), 0.01 uM in saliva (ref 14). A paper not cited by Silva and coworkers measure MEHP in serum (relevant for the in vitro effects) and found that there were not detectable levels of MEHP, but that it was all glucuronidated, and so almost certainly inactive against the receptors. https://link.springer.com/article/10.1007/s00204-003-0486-3

The US EPA has published 18 ER assays and compiled a model of ER activity for agonist and antagonist. See https://academic.oup.com/toxsci/article/148/1/137/1659792 and references therein. This work concluded that MEHP was negative as an agonist or antagonist up to 100 uM.

This group published a second paper on AR with 12 assays, concluding that MEHP was negative as both an agonist and antagonist, again up to 100 uM. See https://pubs.acs.org/doi/abs/10.1021/acs.chemrestox.6b00347 and references therein.

Further papers have reported on ER and/or AR in vitro activity of MEHP : http://www.ijdb.ehu.es/web/paper.php?doi=10.1387/ijdb.092883gs (La Sala et al)

Hershberger (in vivo AR) assays have been run on MEHP:

https://www.tandfonline.com/doi/abs/10.1080/15287390701432285 (Lee et al.)

https://www.sciencedirect.com/science/article/pii/S0300483X04006134 (Stroheker et al) - cited by the authors

https://www.sciencedirect.com/science/article/pii/S0278691506001943 (Stroheker et al)

Uterotrophic evaluation has been carried out

https://europepmc.org/abstract/cba/569868 (Zhang et al)

So in summary, the authors have presented a negative effect when this was alredy well documented, and have largely ignored the literature in the field.

Reviewer 2 Report

In this study, the authors provided the antagonistic effects of MEHP of estrogen and androgen receptors. There are several reports on the anti-estrogenic action of MEHP, as the authors described in the section of discussion. It is also suggested that the properly evaluation of anti-estrogenicity is difficult because MEHP has a strong cytotoxicity to mammalian cells. Therefore, the authors should insist on the advantages of this methodology in the section of introduction.

The authors assert hormone receptors-independent effects of MEHP, but their evidence is low. That ground should be clearly indicated. It is reported that MEHP exerts cell proliferative effects through membrane receptor, GPER signaling. Discussion regarding further mechanisms and signaling of MEHP action should be necessary. 

How were the concentrations of estrogen selected in this research? It is necessary to clarify the concentrations of other compounds as well. References should be provided showing. How many times did you examine each experiment? It is also necessary to check the reproducibility.

Separating the results and discussion will make the results easier to understand.

Reviewer 3 Report

Reviewer critics to MOLECULES-415555 manuscript

In this study, authors investigated the estrogen/androgen and anti-estrogen/androgen activities of mono–(2–ethylhexyl) phthalate (MEHP). The evaluation of this manuscript is easy from biological point of view because the structure of the manuscript is logical and traceable. The whole manuscript is well described and appropriately thorough except the parts of LC-MS/MS.

1; Information about the method performance (LC-MS/MS) should be described in detail, for example MS/MS transitions.

2; To evaluate the manuscript, reviewer must see the chromatograms of the MEHP in control and exposed samples.

3; Where did the next information come from "the decreases of MEHP amount was observed."? Does it based on only profile mass spectrums? Figure 5 represents only strange peak shapes, not contains other information. If it is really LC-MS/MS analysis, the quantification must perform via standard analytical procedure.

4; “2.3 Fate of MEHP in yeast cells” part need to be critically discussed. The Reviewer is not sure that these results are suitable for drawing important conclusions.

5; Authors mentioned the possibility of present of an unknown peak. Discovering or detecting this adduct could provide the really novelty and declare the importance of this part.

6; “4.4 Data analysis”. Is the calculation of induction ratio (IR) a new calculating method?

If yes, please compare with the previous methods.

If not, please refer the previous method or important publications.

Based on these critical issues, reviewer recommend the major revision of the manuscript.